# Effectiveness of Dual Biologic or Small Molecule Therapy for Achieving Endoscopic Remission in Refractory Inflammatory Bowel Disease

**DOI:** 10.3390/diseases10040102

**Published:** 2022-11-09

**Authors:** Israa Abdullah, Khaled AlMojil, Mohammad Shehab

**Affiliations:** 1Department of Pharmacy, Clinical Pharmacy Unit, Kuwait Hospital, Sabah Al-Salem 44001, Kuwait; 2Division of Gastroenterology, Department of Internal Medicine, Mubarak Alkabeer University Hospital, Kuwait University, Jabriya 47060, Kuwait

**Keywords:** IBD, Crohn’s, refractory, biologics, combination, efficacy, safety, dual, target

## Abstract

Inflammatory bowel disease (IBD) is a chronic autoimmune disease with relapse–remission courses. A number of patients may present with a refractory disease with partial or no response to treatment. Others may present with extra-intestinal manifestations that makes the treatment with one biologic challenging. Dual target therapy (DTT), combining biologics and/or small molecule drugs, may offer a chance to achieve remission in these cases and improve patients’ quality of life despite the limited evidence regarding this approach. We present a case series of refractory inflammatory bowel disease cases managed with DTT. Seven patients with refractory IBD achieved steroid free, clinical, and endoscopic remission by using DTT. These results support that DTT could be an effective approach in selected patients with refractory IBD or with concomitant extra-intestinal manifestations (EIM). Larger studies, ideally randomized controlled trials, are needed to further support the evidence and confirm the efficacy and safety of DTT for IBD.

## 1. Introduction

Inflammatory bowel diseases (IBD), including Crohn’s disease (CD) and ulcerative colitis (UC), are chronic autoimmune inflammatory diseases with a relapse–remission nature [1,2]. Biologics and small molecule therapies are considered an effective treatment for moderate–severe cases, whether alone or in combination with an immunomodulator [1,2]. However, several patients fail conventional therapy and present with a refractory disease. Furthermore, a significant proportion of these patients have concurrent extraintestinal manifestations (EIM) which further affect their quality of life and complicate the treatment of the disease [3].

Combining biologic therapies together or with a small molecule, to target different inflammatory pathways, is a new approach with a promising potential for the treatment of patients with refractory disease and those with EIM [4]. A notable number of patients on biologics may not respond to initial therapy (primary non-responders) or may later lose treatment effect (secondary loss of effect). Refractory IBD poses a challenge in treatment where conventional therapy may fail to provide a sufficient response despite treatment optimization. This can limit the treatment options, resulting in poor outcomes, and can adversely affect the patient’s quality of life. Additionally, patients with IBD presented with EIM are more challenging to treat as one medication may not be adequate to control both conditions [3]. In both cases, dual targeted therapy (DTT) offers additional chances for improvement. Anti-TNF drugs are monoclonal antibodies that suppress inflammation by binding to human tumor necrosis factor (TNF) [1,5]. Ustekinumab, on the other hand, interferes with the proinflammatory cytokines and interleukin (IL)-12/IL-23 [1,5]. Vedolizumab binds to α4β7 integrin, thus, inhibiting T-lymphocytes from crossing into gastrointestinal tissues [1,5]. Moreover, tofacitinib is a small molecule of Janus kinase (JAK) inhibitor [2]. Combining biologics or small molecule drugs with different mechanisms of action can help target more than one pathway of inflammation to achieve a synergistic effect. The most common combinations used are anti-TNFs with vedolizumab, and vedolizumab with ustekinumab [6]. The safety profile of these medications, along with their efficacy, have made them an attractive choice for combinations [6].

Large studies documenting the safety and efficacy of DTT are limited. Moreover, randomized controlled trials (RCTs) and long-term follow up of dual biological therapy are deficient, limiting the evidence regarding its safety and efficacy. Two meta-analyses studies reported the efficacy and safety of DTT; however, the efficacy and safety of individual DTT combinations were not reported [4,7]. Thus, we present a case series to report the efficacy and safety of DTT combinations in IBD patients with partial or no response to conventional therapy.

## 2. Case History

### 2.1. Case One

A 33-year-old female presented to the clinic in March 2020 with chronic abdominal pain and diarrhea. Her complete blood count (CBC) showed a hemoglobin of 10.2 g/dL and blood serum C-reactive protein (CRP) of 70 mg/L. Her fecal calprotectin was 823 μg/g. In addition, her colonoscopy showed moderate to severe ileocecal inflammation consistent with Crohn’s disease (CD). Her biopsies confirmed the diagnosis. Her magnetic resonance enterography (MRE) also showed similar findings to her colonoscopy. No fistulas or strictures were seen. Her gastroscopy showed large multiple antral ulcers. The biopsies were consistent with CD, with a presence of granuloma. She was treated initially with prednisone to induce remission. She refused the anti-TNF therapy so she was started on a ustekinumab (STELARA^®^, Janssen Biotech, Horsham, PA, USA) intravenous induction dose followed by a 90 mg subcutaneous injection every 8 weeks. Her diffused abdominal pain resolved over a few months as well as her diarrhea. However, she continued to have epigastric pain despite being on high doses of proton pump inhibitor (PPI).

After 9 months, her CRP was 25 mg/L and fecal calprotectin level was 250 μg/g. A repeated colonoscopy was normal. However, her gastroscopy showed a persistent gastric ulcer and the biopsies were consistent with CD. Her ustekinumab was increased to every 4 weeks. However, 9 months later, her repeated gastroscopy showed similar findings with no improvements. Vedolizumab (ENTYVIO^®^, Takeda Pharmaceuticals; Linz, Austria), a 300 mg intravenous (IV) induction dose, at weeks zero, two, and six, followed by 300 mg IV every 8 weeks, was added to her treatment. After a few months, she started feeling better and her dyspeptic symptoms disappeared. Her Harvey–Bradshaw Index (HBI) score was 3 and her CRP was 4 mg/L. A gastroscopy repeated 12 months after dual therapy showed a resolution of previous gastric ulcers.

### 2.2. Case Two

A 24-year-old male presented in January 2019 with abdominal pain and diarrhea for 3 months. His blood tests showed a hemoglobin of 9.8 g/dL, blood serum C-reactive protein (CRP) of 37 mg/L, and fecal calprotectin of 525 μg/g. The colonoscopy showed terminal ileitis while the gastroscopy showed multiple gastric and duodenal ulcers up to the third part of the duodenum. The biopsies were consistent with Crohn’s disease (CD). His magnetic resonance enterography (MRE) showed 10 cm of distal terminal ileitis. His body mass index (BMI) was 17.5 kg/m^2^. He was started on prednisone oral tapering dose, and on 5 mg/kg of infliximab (REMICADE^®^, Janssen Biotech; Leiden, the Netherlands) and azathioprine for maintenance. He was also kept on high-dose proton pump inhibitor (PPI). A year later, he continued to have epigastric pain and was still losing appetite and unable to gain weight. His BMI was 18.0 kg/m^2^. Repeated laboratory tests showed a hemoglobin of 10.5 g/dL, C-reactive protein of (CRP) 22 mg/L, and fecal calprotectin of 182 μg/g. On the other hand, colonoscopy was normal and MRE showed a resolution of previous ileitis. His gastroscopy, however, showed persistent gastric and duodenal ulcers. Biopsies were consistent with Crohn’s disease. His infliximab trough level was 13 mg/mL, while antibodies were undetectable. He pursued on taking PPI and he was started on an open capsule budesonide for 3 months. However, after 6 months of his gastroscopy findings, he remained the same. He was then started on a 300 mg vedolizumab (ENTYVIO^®^, Takeda Pharmaceuticals; Linz, Austria) intravenous (IV) induction dose at weeks zero, two, and six, followed by 300 mg of IV every 8 weeks. After a couple of weeks, he started to feel better and, 24 weeks later, he was in complete clinical remission, with a HBI score of two. A repeated gastroscopy, 12 months later, showed a complete resolution of the previous gastric and duodenal ulcers. His hemoglobin rose to 13.4 g/dL, CRP decreased to 7 mg/L, and his BMI increased to 19.5 kg/m^2^.

### 2.3. Case Three

A 28-year-old female presented in February 2018 with abdominal pain and bloody diarrhea. Her blood tests showed a hemoglobin of 10.8 g/dL, blood serum C-reactive protein (CRP) of 27 mg/L, and her fecal calprotectin was 621 μg/g. Her colonoscopy showed pancolitis consistent with Mayo 3 and the colonic biopsy confirmed the diagnosis of ulcerative colitis (UC). She was started on a prednisone oral tapering dose and 5-aminosalicylic acid. Unfortunately, her symptoms did not improve, and she continued to have bloody diarrhea. Therefore, she was started on 5 mg/kg of infliximab (REMICADE^®^, Janssen Biotech; Leiden, Netherlands) and azathioprine. In 2019, a repeated colonoscopy showed an active pancolitis consistent with Mayo 2. Her laboratory tests showed a CRP of 20 mg/L and a fecal calprotectin of 450 μg/g. Her stool tests, including

Clostridium difficile (C. diff) and Cytomegalovirus (CMV), were all negative. Her infliximab trough level was 5 mg/mL and infliximab antibodies were positive. She was switched to an adalimumab (HUMIRA^®^, AbbVie Inc.; Ludwigshafen, Germany) induction dose of 160 mg, followed by 80 mg, then by 40 mg every other week in addition to a prednisone oral tapering dose. Three months later, she continued to have bloody diarrhea with a fecal calprotectin of 700 μg/g and CRP of 28 mg/L. A repeated colonoscopy showed active Mayo 3 pancolitis. Her stool tests were negative again. Therefore, she was switched to a 390 mg ustekinumab (STELARA^®^, Janssen Biotech, Horsham, PA, USA) intravenous induction dose, followed by 90 mg every 8 weeks. Initially, she showed an improvement; however, 3 months later, she continued to have diarrhea and abdominal pain. The colonoscopy showed active Mayo 2 pancolitis with negative pathogens. Accordingly, she received a prednisone oral tapering dose and ustekinumab was increased to every 4 weeks. After tapering the prednisone dose, she flared again with bloody diarrhea approximately 10 times a day. Her test results showed a hemoglobin of 9.4 g/dL, CRP of 35 mg/L, and fecal calprotectin of more than 1000 μg/g. She was admitted to the hospital and received intravenous (IV) fluids, IV corticosteroids, and iron infusion. In addition, ustekinumab was discontinued and she was started on a 10 mg tofacitinib (XELJANZ^®^, Pfizer Inc; Freiburg, Germany) twice-daily induction dose orally. On follow up, she stated that she continued to have diarrhea approximately three to four times a day, but without blood. On follow up, her blood serum CRP dropped to 18 mg/L, her fecal calprotectin was 224 μg/g, and stool tests, including C. diff and CMV, were negative. Two months later, with persistent diarrhea and abdominal pain, a colonoscopy showed a Mayo score of 1 colitis in the left side and a Mayo score of 2 colitis in the right side. Compared with the previous colonoscopy, this colonoscopy showed a great improvement; however, she still had active disease. Due to the improvement on tofacitinib, the decision was made to continue tofacitinib and to start her on 300 mg of vedolizumab (ENTYVIO^®^, Takeda Pharmaceuticals; Linz, Austria) IV every 8 weeks along with a prednisone oral tapering dose. After 6 months, she started to feel better and her symptoms resolved, with a partial Mayo score of 1. Moreover, her blood serum CRP decreased to 8 mg/L and her fecal calprotectin to 181 μg/g. A year later, a repeated colonoscopy showed a complete resolution of her colonic disease with a Mayo score of 0.

### 2.4. Case Four

A 32-year-old female was diagnosed in 2010 with severe ileocolonic Crohn’s disease (CD). Her initial blood tests were normal, and fecal calprotectin was 522 μg/g. A colonoscopy showed terminal ileitis. The biopsies were consistent with Crohn’s disease. Her magnetic resonance enterography (MRE) showed 15 cm of terminal ileitis. Initially, she was started on a prednisone oral tapering dose and 40 mg of subcutaneous adalimumab (HUMIRA^®^, AbbVie Inc.; Ludwigshafen, Germany) every 2 weeks. After 2 years, she developed a secondary nonresponse to adalimumab and was switched to infliximab (REMICADE^®^, Janssen Biotech; Leiden, the Netherlands). After being on infliximab for 3 years, in 2017, she flared up and developed secondary nonresponse to infliximab, so she was switched to 400 mg of certolizumab (CIMZIA^®^, UCB Group Inc.; Braine-l’Alleud, Belgium) every 4 weeks. In 2018, she was admitted with small bowel obstruction. A computed tomography scan (CT scan) showed a 7 cm stricture in the distal terminal ileum. Her blood serum C-reactive protein (CRP) was 45 mg/L and fecal calprotectin was 750 μg/g. She had a small bowel resection as she failed conservative medical therapy. Postoperatively, she did well and was discharged on metronidazole for 3 months. A repeated colonoscopy, after 6 months, showed newly developing ulcers in the new terminal ileum with a Rutgeerts score i4. Because of these new lesions, she was started on 90 mg of ustekinumab (STELARA^®^, Janssen Biotech, Horsham, PA, USA) every 8 weeks. Nine months later, a repeated colonoscopy showed no change on those ulcers with the same Rutgeerts score of i4 and MRE showed an active terminal ileitis of 10 cm. Accordingly, her ustekinumab dose frequency was increased to every 4 weeks.

A year later, with mild symptoms and diarrhea three to four times per day, colonoscopy showed resolution of her terminal ileum ulcer; however, MRE showed patchy jejunal inflammation. Her repeated laboratory tests showed a normal complete blood count, blood serum CRP of 15 mg/L, and fecal calprotectin of 400 μg/g. Therefore, 300 mg of vedolizumab (ENTYVIO^®^, Takeda Pharmaceuticals; Linz, Austria) intravenous every 8 weeks was added to her treatment regimen. On follow up, 1 year later, she was asymptomatic and doing well. Her HBI score was 3, blood serum CRP was 5 mg/L, fecal calprotectin was 112 μg/g, and her colonoscopy and MRE showed a complete resolution of her ileitis with no residual disease.

### 2.5. Case Five

A 20-year-old female was diagnosed in 2019 with ileocolonic and gastric Crohn’s disease (CD). A colonoscopy showed ileocolic valve stenosis and gastroscopy showed duodenal ulcers. The biopsies were consistent with CD. Her magnetic resonance enterography (MRE) showed 10 cm of terminal ileitis. Initially, she was started on prednisone orally as a tapering dose and a 260 mg ustekinumab (STELARA^®^, Janssen Biotech, Horsham, PA, USA) intravenous induction dose, followed by 90 mg every 8 weeks. Nine months later, a repeated colonoscopy and gastroscopy showed a complete resolution of her duodenal ulcers; however, her terminal ileitis and inflammatory ileocecal stenosis were still present. Due to that, her ustekinumab dose frequency was increased to every 4 weeks. Unfortunately, 3 months later, she presented to the emergency department with a small bowel obstruction. The colonoscopy, 6 months postoperatively, showed new ulcers consistent with Rutgeerts i4, and accordingly she was started on 5 mg/kg of intravenous infliximab (REMICADE^®^, Janssen Biotech; Leiden, the Netherlands).

Six months later, she presented with abdominal pain and diarrhea. Her blood serum C-reactive protein (CRP) was 40 mg/L and fecal calprotectin was 921 μg/g. A colonoscopy showed new terminal ileum ulcers with a persistent Rutgeerts score of i4. Her infliximab trough level was 3 mg/mL and antibodies were positive. Therefore, she was switched to a 160 mg adalimumab (HUMIRA^®^, AbbVie Inc.; Ludwigshafen, Germany) subcutaneous induction dose, followed by 80 mg then 40 mg every 2 weeks. A year later, she presented with a second episode of small bowel obstruction and had undergone another surgery due to stenotic terminal ileum. Two weeks postoperatively, she was switched to 400 mg of certolizumab (CIMZIA^®^, UCB Group Inc.; Braine-l’Alleud, Belgium) every 4 weeks. On follow up, 6 months later, the colonoscopy showed no new terminal ileum ulcers, and her tests showed a blood serum CRP of 8 mg/L and fecal calprotectin of 88 μg/g.

After 9 months on certolizumab, she developed severe acid reflux. A repeated colonoscopy was normal. However, gastroscopy showed new esophageal ulcers consistent with CD confirmed by a biopsy. She was started on an oral proton pump inhibitor, prednisone oral tapering dose, and 300 mg of vedolizumab (ENTYVIO^®^, Takeda Pharmaceuticals; Linz, Austria) IV every 8 weeks in addition to certolizumab. After a couple of weeks, she started to feel better and, after 6 months, she was in complete clinical remission. A year later, her gastroscopy and colonoscopy showed a complete resolution of the esophageal ulcers, with no signs of active CD, and an HBI score of 2. MRE was also normal. Her blood serum CRP decreased to 4 mg/L and fecal calprotectin decreased to 52 μg/g.

### 2.6. Case Six

A 35-year-old female presented in 2015 with abdominal pain and diarrhea. Her initial C-reactive protein (CRP) was 41 mg/L and fecal calprotectin was 954 μg/g. A colonoscopy showed ileocecal valve stenosis, inflammatory ulcers, and terminal ileitis, but her gastroscopy was normal. Her biopsies were consistent with Crohn’s disease (CD). Her magnetic resonance enterography (MRE) showed 15 cm of terminal ileitis with ileocecal valve stenosis. Initially, she was started on a prednisone oral tapering dose and 40 mg of subcutaneous adalimumab (HUMIRA^®^, AbbVie Inc.; Ludwigshafen, Germany) every 2 weeks. On follow up, 6 months later, she started complaining from abdominal pain and diarrhea again. A repeated colonoscopy and MRE showed similar findings to the previous results. At that time, her blood serum CRP was 30 mg/L and fecal calprotectin was 621 μg/g. Her adalimumab trough level was 8 mg/mL and antibodies were undetectable. Therefore, she was switched to a 390 mg ustekinumab (STELARA^®^, Janssen Biotech, Horsham, PA, USA) intravenous induction dose followed by 90 mg every 8 weeks.

After being on ustekinumab for 1 year, her colonoscopy showed a mild improvement of the previous ulcers; however, she continued to have ileocecal valve stenosis. Her MRE showed a partial resolution of her ileitis with persistent 5 cm of ileitis. For this reason, she received prednisone orally as a tapering dose and her ustekinumab dose was increased to every 4 weeks. Nine months later, her blood serum CRP was 22 mg/L and fecal calprotectin was 521 μg/g. Her gastroscopy was normal, and her colonoscopy showed a resolution of the ileocecal ulcerations without ileitis. However, her MRE showed new patchy proximal ileitis with a resolution of the previous distal ileitis; hence, 300 mg of intravenous vedolizumab (ENTYVIO^®^, Takeda Pharmaceuticals; Linz, Austria) every 8 weeks was added to her treatment. After a year, she started feeling better and her symptoms disappeared, with a HBI score of 4. Her repeated colonoscopy continued to be normal and her MRE showed a resolution of the new proximal and previous distal ileitis. Her blood serum CRP decreased to 8 mg/L and fecal calprotectin to 121 μg/g.

### 2.7. Case Seven

A 42-year-old male presented in 2015 with acute abdominal pain and vomiting and was diagnosed with a small bowel obstruction based on his computed tomography (CT) scan. His CT scan showed long segment ileitis with a short fibrotic stricture at the ileocecal valve. He also had enteroenteric fistula. He failed steroids and conservative treatment, so he had small bowel resection fistulectomy and right hemicolectomy. He did well and was discharged home. Pathology confirmed penetrating Crohn’s disease. Three months later, he started to have diarrhea and abdominal pain. His blood serum C-reactive protein (CRP) was 42 mg/L and fecal calprotectin was 544 μg/g. The rest of his labrotory tests, including stool tests, were unremarkable. His colonoscopy showed neo-terminal ileum severe ulcerations along with severe pan-colonic ulcerations. His MRE showed 7 cm terminal ileitis. No strictures were seen. His MRE pelvis was normal. He was started on a 5 mg/kg infliximab (REMICADE^®^, Janssen Biotech; Leiden, The Netherlands) induction dose, follow by a maintenance dose, along with azathioprine 2 mg/kg. He did well for 5 years and was in clinical and endoscopic remission.

In 2020, he developed another flare with abdominal pain and diarrhea. His blood serum CRP was 34 mg/L and fecal calprotectin was 721 μg/g. A colonoscopy showed severe ileocolonic disease. His infliximab trough level was 2 mg/mL and antibodies to infliximab were detected. He was started on an adalimumab (HUMIRA^®^, AbbVie Inc.; Ludwigshafen, Germany) induction dose followed by a 40 mg subcutaneous (S.C) injection every 2 weeks for maintenance. He was having a partial response so his adalimumab was increased to 80 mg S.C every week over 12 months. He then felt better but he was complaining of three to four bowel motions per day and occasional abdominal pain. A repeated colonoscopy showed a resolution of the previous neo-terminal ileal disease but persistent colonic disease. A repeated MRE also showed a normal small bowel with no active disease or fistula formation. His adalimumab trough level was 12 mg/mL. Vedolizumab (ENTYVIO^®^, Takeda Pharmaceuticals; Linz, Austria) standard induction and maintenance therapy was added. In 6 months, he was feeling well with no symptoms, and had a HBI score of 3. His blood serum CRP was 7 μg/g and his fecal calprotectin was 123 μg/g. A colonoscopy 12 months later was completely normal with no signs of active disease. A summary of the cases is presented in Table 1.

## 3. Discussion

We presented seven cases of patients with refractory inflammatory bowel disease (IBD), including cases with ulcerative colitis and Crohn’s disease, who successfully achieved clinical and endoscopic remission following dual biologic therapy. These cases support the efficacy of combining biologic therapy in patients with partial or no response to conventional IBD treatment options. Currently, data supporting the routine use of dual biologic therapy in IBD patients are limited and randomized controlled trials (RCT) are lacking, except for a single RCT involving natalizumab with an anti-TNF [8].

Dual targeted therapy (DTT) seems to offer a better chance for improvement with several studies supporting that finding [7,8,9,10,11,12,13]. Most combinations used are vedolizumab, tofacitinib, anti-TNFs, and ustekinumab due to their considerably safer profile and varying mechanism pathways. The safety of DTT has also been explored in many of the studies with few reporting no serious adverse events [8,9] and others reporting few cases of increased risk of infections [7,10,11,13]. Our case report supports the finding of these studies with an improvement in clinical outcomes and no serious adverse events reported.

To date, only one RCT included DTT, with natalizumab, a monoclonal antibody, that currently has very limited use in IBD treatment due to its association with progressive multifocal leukoencephalopathy [8]. The double-blinded RCT investigated the safety and tolerability of natalizumab combined with infliximab in patients with refractory Crohn’s disease [8]. Patients on combination therapy had similar rates of adverse events, compared with the placebo group, with no opportunistic infections observed [8]. As a secondary outcome, the combination was also found more effective in achieving clinical improvement, even though the study was not powered to detect statistical significance in efficacy [8]. Moreover, a meta-analysis by Ribaldone et al. included seven studies with a total of 18 patients on DTT of vedolizumab with an anti-TNF or ustekinumab and approximately 14 months of follow up. All patients (100%) in the study achieved clinical improvement and 93% had endoscopic improvement. None of the adverse events reported were serious, favoring the use of the DTT in refractory IBD cases [7]. A more recent meta-analysis examined the safety and efficacy of DTT, biologic or small molecule therapy, in 30 studies. A total of 279 patients with refractory IBD or with extra-intestinal symptoms were involved, of which 59% achieved clinical remission and 34% reached endoscopic remission. Serious adverse events, mostly infections, were 6.5% and mainly related to outcomes from one study. The variety of combinations with a limited number of studies also adds a limitation to the above study [4].

Several cohorts [9,10,11,13] investigated the approach of DTT in IBD refractory cases and in IBD patients with concomitant autoimmune disease. Glasner et al., Kwapisz et al., Llano et al., Privitera et al., and and Yang et al. all found an improvement in clinical outcomes in the refractory cases or in patients with extraintestinal manifestations (EIM) when dual biologics were used [9,10,11,13]. In terms of safety, Privitetera et al. reported few adverse events, none of which were severe, while Glasner et al., Ribaldone et al., Yang et al., and Kwapisz et al. reported few serious adverse events, most of which were related to an increased risk of infection [9,10,11,13]. Additionally, a multicenter retrospective cohort examined the efficacy and safety of combining biologics together or with a small molecule. The combinations resulted in a clinical and endoscopic improvement in more than 50% of IBD patients and a better control of the extraintestinal manifestations. The study, however, reported a significant number of adverse events (42%) and an increased risk of infections, which resulted in 10% hospitalization [14].

Furthermore, a case series by Biscaglia et al. found clinical improvement in IBD symptoms and EIM, while no adverse events were reported post the 2 years follow up with DTT [15]. Similarly, most of the cases reported by Moe et al., after a follow up of 5–37 months, achieved clinical remission or had an improvement in EIM symptoms with few incidents of non-severe infections [16]. A review by Hirten et al. discussed the results of one RCT, by Sands et al., two case reports, and a case series of which all reported improvement in IBD symptoms in the refractory cases with the dual biologics in the cases of refractory IBD and those with IBD and concurrent extraintestinal manifestations not controlled on one agent [17]. Moreover, Yzet et al. reported three cases of IBD with paradoxical psoriasis treated with an anti-TNF and ustekinomab. All the three cases in the study had a controlled IBD but resistant psoriasis that did not show improvement despite being on dual biologics. No adverse events were reported. The study, however, did not report details regarding the severity of the lesion and drugs frequency, and the treatment was limited to combinations of infliximab or adalimumab with ustekinomab [18].

Further studies are needed to examine the long-term safety and efficacy of the DTT in these groups of patients. Interestingly, there is an ongoing phase 2a double-blind RCT evaluating the safety and efficacy of guselkumab with golimumab in moderate to severe UC [19]. At 12 weeks, patients on DTT achieved more clinical remission and endoscopic improvement, while adverse events were similar among all groups [19]. Another phase 4 clinical trial, the EXPLORER, combining adalimumab, vedolizumab, and methotrexate is currently ongoing (NCT02764762, clinicaltrials.gov). Hopefully, the results of those trials will further strengthen the evidence regarding dual therapy to target different autoimmune pathways concomitantly.

The results of this study support the efficacy and safety of DTT for refractory cases where conventional therapy does not provide sufficient control of the disease. Our case series included patients on various DTT combinations of biologics and small molecule drugs (tofacitinib). We included all the patients in our center who were on DTT, and they were followed for a minimum of 12 months. The use of DTT seems to be considerably safe. During the whole follow-up period, none of the patients in our study developed any serious infection or adverse events. Moreover, all patients in our center with refractory IBD whose symptoms could not be controlled with conventional IBD treatment and who were placed on DTT showed a significant clinical and endoscopic improvement. These results will provide more treatment options to clinicians and a better chance to control refractory cases of IBD. Patients whose symptoms could not be sufficiently controlled with conventional therapy would also have hope to have an improved quality of life.

## 4. Conclusions

Refractory inflammatory bowel disease (IBD) cases and patients with coexisting extraintestinal manifestations (EIM) poses a challenge in the treatment and can negatively affect patients’ quality of life. Targeting different immunological pathways to control the disease through combining biologics may offer a chance for those patients to reach remission. Even though there is not enough current evidence to support the routine use of dual targeted therapy (DTT) in IBD, weighing the risk to benefits in patients with refractory IBD and patients with concurrent EIM may grant the use of DTT in this group of patients. Our case series adds to the evidence that DTT can be effective with low risk of severe infection.

## Figures and Tables

**Table 1 diseases-10-00102-t001:** Summary of Patients’ Characteristics and Dual Targeted Therapy Used.

Characteristics	Sex	Age	Type of IBD	Dual Targeted Therapy Used	Clinical Remission (HBI/Partial Mayo) Score ^+^	Endoscopic Remission (SES-CD/Mayo) Score *	Time to Endoscopic Remission (Months)
Case 1	Female	33	Crohn’s disease	Ustekinumab + vedolizumab	3	1	12 months
Case 2	Male	34	Crohn’s disease	infliximab + vedolizumab	2	2	12 months
Case 3	Female	28	Ulcerative colitis	tofacitinib + vedolizumab	1	0	12 months
Case 4	Female	32	Crohn’s disease	ustekinumab + vedolizumab	3	2	12 months
Case 5	Female	20	Crohn’s disease	certolizumab + vedolizumab	2	1	12 months
Case 6	Female	35	Crohn’s disease	ustekinumab + vedolizumab	4	1	12 months
Case 7	Male	41	Crohn’s Disease	Adalimumab + vedolizumab	3	1	12 months

IBD: inflammatory bowel disease; UC: ulcerative colitis; CD: Crohn’s disease; HBI: Harvey–Bradshaw Index; SES-CD: Simple Endoscopic Score for Crohn’s Disease. ^+^ Normal range clinical score: partial Mayo score < 2; HBI < 5. * Normal range endoscopic score: UC Mayo score 0; Crohn’s disease SES-CD 0-2.

## Data Availability

Not applicable.

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
