# Peer review of "Effectiveness of Dual Biologic or Small Molecule Therapy for Achieving Endoscopic Remission in Refractory Inflammatory Bowel Disease"

_diseases, 2022, doi:10.3390/diseases10040102_

Round 1

Reviewer 1 Report

Manuscript # diseases-1901158

Title: Effectiveness of Dual Biologic or Small Molecule Therapy for Achieving Endoscopic Remission in Refractory Inflammatory Bowel Disease

Authors: Abdullah et al. 

The reason for preparing of above manuscript is unclear. The manuscript presents 7 clinical cases showing that the use of vedolizumab in combination with ustekinumab, infliximab, tofacitinib, certolizumab, or adalimumab is effective in achieving a remission in patients with refractory IBD. It should be noted, however, that beneficial effects of combination therapy in IBD have already been shown in randomized controlled trails and also presented as a recommendation in the AGA Clinical Practice Guidelines. Therefore, it is incomprehensible to present case reports as a support of these recommendations. The more that it is not known whether the authors present all cases that they observed, or only those that confirm their concept. Therefore, the manuscript should be rejected. In addition, it should be noted that the manuscript contains many other errors. The language of the manuscript is incomprehensible, and one has to guess what authors meant. In some cases, the form is so ambiguous that it cannot be interpretated unambiguously. There are numerous typing errors, some abbreviations are without explaining their meaning and are incorrectly used. The case reports are incomplete, there is no data on the trade name of medicine used, as well as name of manufacturer, city, and country. The doses of drugs used were not given.

Author Response

The reason for preparing of above manuscript is unclear. The manuscript presents 7 clinical cases showing that the use of vedolizumab in combination with ustekinumab, infliximab, tofacitinib, certolizumab, or adalimumab is effective in achieving a remission in patients with refractory IBD. It should be noted, however, that beneficial effects of combination therapy in IBD have already been shown in randomized controlled trails and also presented as a recommendation in the AGA Clinical Practice Guidelines. Therefore, it is incomprehensible to present case reports as a support of these recommendations. The more that it is not known whether the authors present all cases that they observed, or only those that confirm their concept. Therefore, the manuscript should be rejected. In addition, it should be noted that the manuscript contains many other errors. The language of the manuscript is incomprehensible, and one has to guess what authors meant. In some cases, the form is so ambiguous that it cannot be interpretated unambiguously. There are numerous typing errors, some abbreviations are without explaining their meaning and are incorrectly used. The case reports are incomplete, there is no data on the trade name of medicine used, as well as name of manufacturer, city, and country. The doses of drugs used were not given.

  • Thank you for the review. The aim of the study was rephrased to clarify the goal of this report. Please see line 59.
  • The aim of this study is to investigate the efficacy and safety of dual targeted therapy (DTT) that includes the use of two biologics or a biologic combined with a small molecule drug for refractory cases of inflammatory bowel disease. RTC in this matter are limited to only one old  published and a few ongoing trials (Sands BE et al, 2007). The beneficial effect of the combination has been a place of discussion with some studies reporting improvement and questioning the risk of infection. Please see discussion section.
  • A sentence was added under methods section: All the cases in our center with refractory IBD who have been placed on dual targeted therapy (DTT) were included in this study. Thank you.
  • Manuscript revised, typing errors and abbreviations corrected.
  • All the biologic medications used in the study were the original approved biologics (originator)- no biosimilars were used. However, to avoid any ethical or marketing issues, we use only generic names through-out the manuscript. Most similar studies as well uses generic names only without specifying the trade name used. Examples are a study by Glassner K et al and a study by sandborn WJ et al (see references below). The city, country, trade names and other details don’t have any influence on the study and should not affect the results.

  1. Glassner K, Oglat A, Duran A, Koduru P, Perry C, Wilhite A, Abraham BP. The use of combination biological or small molecule therapy in inflammatory bowel disease: a retrospective cohort study. Journal of Digestive Diseases. 2020 May;21(5):264-71.
  2. Sandborn WJ, Feagan BG, Rutgeerts P, Hanauer S, Colombel JF, Sands BE, Lukas M, Fedorak RN, Lee S, Bressler B, Fox I. Vedolizumab as induction and maintenance therapy for Crohn's disease. New England Journal of Medicine. 2013 Aug 22;369(8):711-21.

Reviewer 2 Report

The author has presented the article in a fantastic way. It explains the importance of Dual Target Therapy that could be potential treatment of people suffering from inflammatory diseases. I have a few points to add that would improve the clarity of the paper,

1. It would be great to know the scores i.e Clinical and Endoscopic remission scores of the patients with single therapy so it would be clear to understand the effect of dual treatment. 

2. Also, it would be great to know if there were patients who had remission after 12 months. This would help understand the effectiveness of dual therapy. 

3. The author could include the remission scores in between therapies since it would help decide whether single therapy would have been effective.

Author Response

The author has presented the article in a fantastic way. It explains the importance of Dual Target Therapy that could be potential treatment of people suffering from inflammatory diseases. I have a few points to add that would improve the clarity of the paper,

  1. It would be great to know the scores i.e Clinical and Endoscopic remission scores of the patients with single therapy so it would be clear to understand the effect of dual treatment. 

Thank you. In this case series we are reporting cases with refractory inflammatory bowel disease who failed single biologic therapy and showed an improvement on the dual target therapy rather than comparing the DTT to single therapy. The study is not a case control study.

  1. Also, it would be great to know if there were patients who had remission after 12 months. This would help understand the effectiveness of dual therapy. 

For this report, all patients have been followed up for up to a year now. In a follow up study we will see the effect and possible safety issues after 12 months.

  1. The author could include the remission scores in between therapies since it would help decide whether single therapy would have been effective.

Thank you. Remission scores for single therapy were not obtained for this study. Since this is a case series, we aimed at reporting our results of refractory cases who responded well to dual target therapy (DTT) with biologics/small molecule drug rather than comparing single therapy to DTT. Thus, the study does not include a control for comparison between single therapy and DTT.

Reviewer 3 Report

The manuscript entitled “Effectiveness of Dual Biologic or Small Molecule Therapy for Achieving Endoscopic Remission in Refractory Inflammatory Bowel Disease” addresses the beneficial clinical outcomes of using dual biologic therapy for triggering endoscopic remission in patients with refractory IBD. Herein, the authors showed 7 clinical cases with refractory IBD that achieved steroid-free, clinical, and endoscopic remission using dual-target therapy.  

Comments:     

1) What is the novelty of the present work? The authors are advised to elaborate in the introduction section on the sharp differences that highlight the novelty of the present work and how the study is different from previous literature that has already addressed the use of dual therapy for IBD. What is already described by the authors regarding this issue in lines 34-36 needs to be further elaborated.

For example, the authors are advised to elaborate on how the current investigation is different from the listed reference 4 which already documents the efficacy of dual therapy for IBD (Ahmed W, Galati J, Kumar A, Christos PJ, Longman R, Lukin DJ, Scherl E, Battat R. Dual biologic or small molecule therapy for treatment of inflammatory bowel disease: a systematic review and meta-analysis. Clinical Gastroenterology and Hepatology. 2021 Mar 31).

2) The aim of the study should be formulated more clearly.

3) The current title of the present work needs to be modified. Since all the presented cases in the present manuscript were treated with dual biologic therapy none of them were managed using small molecule therapy, the authors are advised to remove “small molecule”.

4) From a mechanistic perspective, the authors are advised to elaborate in the introduction on how dual-target therapy is beneficial for the treatment of patients with refractory disease and those with EIM.

5) The authors should discuss the potential clinical significance of the results obtained.

6) To avoid readers’ confusion, table 1 needs to be modified where the range of scores needs to be incorporated into the table text. This would make it clearer for readers to judge the clinical severity of the assessed outcome.  

7) Regarding the institutional board statement “The study protocol was reviewed and approved by the standing committee for coordination of health and medical research at the Ministry of Health in Kuwait (IRB 308 2021/3613)”. Please add the date of the IRB. Apparently, the IRB was issued in 2021, however, the cases were investigated in 2020. Kindly double-check this issue.

8) The manuscript needs to be carefully checked by a native English speaker for grammar and typos. An example is:

- In lines 21-22, the authors state that “Llarger studies, ideally randomized controlled trials, are needed to further 21 support the evidence and confirm efficacy and safety of DTT for IBD”.

Please, replace “Llarger” with “Larger”.

- In line 215: “In 2020, he eveloped another falre with abdominal pain and diarrhea”.

Please correct to “developed” and “flare”, respectively. 

Author Response

1) What is the novelty of the present work? The authors are advised to elaborate in the introduction section on the sharp differences that highlight the novelty of the present work and how the study is different from previous literature that has already addressed the use of dual therapy for IBD. What is already described by the authors regarding this issue in lines 34-36 needs to be further elaborated.

34-35: Combining biologic therapies together or with a small molecule, targeting different inflammatory pathways, is a new approach with a promising potential for the treatment of patients with refractory disease and those with EIM

For example, the authors are advised to elaborate on how the current investigation is different from the listed reference 4 which already documents the efficacy of dual therapy for IBD (Ahmed W, Galati J, Kumar A, Christos PJ, Longman R, Lukin DJ, Scherl E, Battat R. Dual biologic or small molecule therapy for treatment of inflammatory bowel disease: a systematic review and meta-analysis. Clinical Gastroenterology and Hepatology. 2021 Mar 31).

Thank you. Done (please see lines 36-60 under introduction section)

2) The aim of the study should be formulated more clearly.

Aim is rephrased to: To report the efficacy and safety of dual targeted therapy (DTT) in patients with partial or no response to conventional therapy (see line 59). Thank you.

3) The current title of the present work needs to be modified. Since all the presented cases in the present manuscript were treated with dual biologic therapy none of them were managed using small molecule therapy, the authors are advised to remove “small molecule”.

Thank you. The title included biologics and small molecule to cover all the classes of medications included in our case series. In Case 3, the patient was managed with vedolizumab and tofacitinib. Tofacitinib is a small molecule of JAK inhibitor.

4) From a mechanistic perspective, the authors are advised to elaborate in the introduction on how dual-target therapy is beneficial for the treatment of patients with refractory disease and those with EIM.

Thank you. Introduction improved. Please see lines 36-60.

5) The authors should discuss the potential clinical significance of the results obtained.

Thank you. We added additional paragraph to the discussion section. Please see lines 320-331.

6) To avoid readers’ confusion, table 1 needs to be modified where the range of scores needs to be incorporated into the table text. This would make it clearer for readers to judge the clinical severity of the assessed outcome.  

Thank you. A footer is added to the table to include the following:

IBD: inflammatory bowel disease; UC: ulcerative colitis; CD: Crohn’s disease; HBI: Harvey-Bradshaw Index; SES-CD: Simple Endoscopic Score for Crohn Disease

Normal range endoscopic score: UC Mayo score 0; Crohn’s disease SES-CD 0-2.

Normal range clinical score: partial Mayo score<2; HBI<5.

7) Regarding the institutional board statement “The study protocol was reviewed and approved by the standing committee for coordination of health and medical research at the Ministry of Health in Kuwait (IRB 308 2021/3613)”. Please add the date of the IRB. Apparently, the IRB was issued in 2021, however, the cases were investigated in 2020. Kindly double-check this issue.

Data were collected for the study in 2022, after obtaining ethical approval. Patients included in the study were already on dual targeted therapy. A written consent was obtained from the patients prior to data collection as well.

8) The manuscript needs to be carefully checked by a native English speaker for grammar and typos. An example is:

- In lines 21-22, the authors state that “Llarger studies, ideally randomized controlled trials, are needed to further 21 support the evidence and confirm efficacy and safety of DTT for IBD”.

Please, replace “Llarger” with “Larger”.

- In line 215: “In 2020, he eveloped another falre with abdominal pain and diarrhea”.

Please correct to “developed” and “flare”, respectively. 

Thank you. Manuscript  extensively checked and errors are corrected.

Round 2

Reviewer 1 Report

Manuscript # diseases-1901158

Title: Effectiveness of Dual Biologic or Small Molecule Therapy for Achieving Endoscopic Remission in Refractory Inflammatory Bowel Disease

Authors: Abdullah et al.

The manuscript covers a series of 7 case reports. Publication in the form of case report is used when unusual and unexpected interaction between different diseases is observed, for the presentation of unknown diseases or disease symptoms, as well as in the case of unexpected therapeutic effects or unknown side effects of treatment used. The case reports presented by the authors indicate that the use of vedolizumab in combination with ustekinumab, infliximab, tofacitinib, certolizumab, or adalimumab was effective in achieving a remission in patients with refractory IBD. It should be noted, however, that beneficial effects of combination therapy in IBD have already been shown in trails and also presented as a recommendation in the AGA Clinical Practice Guidelines. In the new version of the manuscript, the authors presented the reason for preparing their manuscript and added some information on previous papers in the manuscript topic. Those changes could be accepted. However, the authors should mentioned in their manuscript some important studies by Sands et al. (PMID: 17206633), Hirten et al. (PMID: 26673509), and Yzet et al. (PMID: 27151127). Moreover, they should present a large case series gathering cases from numerous European centers (PMID: 34694746).

In addition to the main objection to the manuscript presented above, there are still errors in relation to which the authors declared that they have been removed: 

  1. What does “stool fecal calprotectin” mean (lines 156-157, 191)? What does “proton pompo” mean (line 75)?
  2. Some abbreviations are without explaining their meaning and are incorrectly used. For example, an abbreviation IBD should be introduced in the first line of abstract. What does DDT mean (line 19)? Abbreviation presented in the abstract should be repeated w in the body of the manuscript. What does CBC mean (line 65)? CRP is not included in CBC testing (line 65). Where was CRP determined? What does MR mean (line 174)?
  3. There is no data on the trade name of medicine used, as well as name of manufacturer, city, and country. The doses of drugs used were not given. The authors answered that to avoid any ethical or marketing issues, we use only generic names through-out the manuscript. However, the effects of drugs and their bioavailability may vary depending on the source of their origin. For this reason, the authors must provide the generic and trade name of medicine used, as well as name of manufacturer, city, and country. The doses of drugs and frequency of their administration are still unknown.

Round 3

Reviewer 1 Report

The manuscript is almost ready for publication. There are still some inaccuracies. It is still unknown in which body fluids CRP was determined. Regarding the drugs used, the authors have provided the trade names and names of manufacturers in the current version of the manuscript but the names of cities and countries where the drugs were produced are still unknown.

Author Response

Please see edited manuscript. Thank you.